# Evaluation of the HearWell Pilot Program: A Participatory *Total Worker Health^®^* Approach to Hearing Conservation

**DOI:** 10.3390/ijerph18189529

**Published:** 2021-09-10

**Authors:** Jennifer M. Cavallari, Adekemi O. Suleiman, Jennifer L. Garza, Sara Namazi, Alicia G. Dugan, Robert A. Henning, Laura Punnett

**Affiliations:** 1Department of Public Health Sciences, UConn School of Medicine, Farmington, CT 06030, USA; asuleiman@uchc.edu; 2Division of Occupational and Environmental Medicine, UConn School of Medicine, Farmington, CT 06030, USA; garza@uchc.edu (J.L.G.); adugan@uchc.edu (A.G.D.); 3Department of Health Sciences, Springfield College, Springfield, MA 01109, USA; snamazi@springfieldcollege.edu; 4Department of Psychological Sciences, University of Connecticut, Storrs, CT 06269, USA; robert.henning@uconn.edu; 5Department of Biomedical Engineering, University of Massachusetts Lowell, Lowell, MA 01854, USA; Laura_Punnett@uml.edu

**Keywords:** *Total Worker Health*, hearing conservation, hearing protection device, safety climate

## Abstract

Our objective was to pilot test HearWell, an intervention created to preserve hearing among highway maintainers, by using a participatory *Total Worker Health^®^* (TWH) approach to designing, implementing and evaluating interventions. Regional maintenance garages were randomized to control (*n* = 6); HearWell (*n* = 4) or HearWell Design Team (*n* = 2) arms. Maintainer representatives from the HearWell Design Team garages identified barriers to hearing health and collaborated to design interventions including a safety leadership training for managers, a noise hazard management scheme to identify noise levels and indicate the hearing protection device (HPD) needed, and a comprehensive HearWell training video and protocol. These worker-designed interventions, after manager input, were delivered to the HearWell Design Team and the HearWell garages. Control garages received standard industry hearing conservation training. Periodic surveys of workers in all 12 garages collected information on the frequency of HPD use and a new hearing climate measure to evaluate changes in behaviors and attitudes over the study period and following interventions. An intention-to-treat approach was utilized; differences and trends in group HPD use and hearing climate were analyzed using a mixed-effects model to account for repeated measures from individual participants. The HearWell Design Team maintainers reported the highest frequency of HPD use. Hearing climate improved in each group 6 months following intervention implementation, with the largest increase and highest value for the HearWell Design Team workers. The HearWell pilot intervention showed promising results in improving HPD use through a participatory TWH approach to hearing conservation. Furthermore, results suggest that employee participation in hearing conservation programs may be necessary for maximal effectiveness.

## 1. Introduction

Despite over 30 years of occupational noise regulation in the United States (US), noise-induced hearing loss (NIHL) remains one of the most common self-reported occupational illnesses or injuries [1]. While identifying the exact number of noise-exposed workers in the US is difficult, it is estimated that approximately 22 million workers are exposed to high levels of noise at work with approximately 25% of all workers reporting hazardous workplace noise exposure at some point in their career [2,3]. Among workers with a history of noise exposure, approximately one-third have NIHL documented through audiometric evaluation and 16% have hearing impairment [4]. In addition to NIHL, high noise exposures are associated with myriad health effects including tinnitus, cardiovascular disease, and sleep disturbances [4]. At the workplace, high noise levels and hearing loss contribute to communication difficulties, safety concerns, and poor job performance [4]. Diminished ability to hear is a disability in itself; hearing loss also greatly impacts quality of life and is strongly associated with depression [5].

While noise exposures were recognized as an occupational hazard for centuries, in 1983 the US—through the Occupational Safety and Health Administration (OSHA) Hearing Conservation Amendment–required hearing conservation programs (HCP) for worksites where noise exposures meet or exceed 85 dBA over an 8-h time-weighted average (TWA) [6]. The OSHA Hearing Conservation Amendment mandates five key components of an HCP: annual audiometry, noise monitoring and exposure control, employee training, provision of hearing protection devices (HPD), and recordkeeping. However, a systematic review indicates a paucity of rigorous research demonstrating the effectiveness of HCP on reducing NIHL [7]. Thus, it remains unclear whether the current burden of NIHL is a reflection of the lack of HCP among noise-exposed workers or the ineffectiveness of HCP in preventing NIHL. Research suggests that certain elements of an HCP appear to be key factors in achieving or failing to achieve program effectiveness, such as management commitment and counseling of workers about NIHL [8].

OSHA’s Hearing Conservation amendment is a performance standard, where the content of the HCP is prescribed, and workplaces may use methods of their choosing to implement their HCP. One approach to addressing worker health is through active employee engagement, a cornerstone of participatory ergonomics and the *Total Worker Health^®^* (TWH) approach, where workers use their expertise to design or change their workplace, work organization and/or jobs to improve physical and psychosocial working conditions [9,10]. In fact, many argue that employee engagement is a core component of an integrated TWH program [11]. Furthermore, an integrated approach to address worker health, safety and well-being, consistent with the TWH approach, was suggested by the National Institute for Occupational Safety and Health (NIOSH) as paramount in addressing the challenges put forth by future work issues [12].

The Center for the Promotion of Health in the New England Workplace (CPH-NEW), a NIOSH TWH Center of Excellence, has designed the Healthy Workplace Participatory Program (HWPP) as a TWH approach for addressing worker health, safety and well-being issues and concerns in which employee engagement is a cornerstone. At the core of the HWPP is the active participation of employees through worker teams and a structured root-cause analysis approach using the Intervention Design and Analysis Scorecard (IDEAS) Tool [13]. The IDEAS approach builds interventions along with a business case for the importance of the worker health, safety and well-being issues that are being addressed. This approach was used in a variety of workplace settings including corrections [14], real estate management [15], state agencies [15], healthcare [16], and retail workplaces [17] to address a variety of health, safety and well-being concerns: burnout [15], blood pressure [17], healthy weight [17,18], work overload [19], sleep [20], and indoor air quality [11].

Our objective was to design, implement and evaluate elements of an HCP, called “HearWell,” using a participatory TWH approach based on CPH-NEW’s HWPP. While the IDEAS process can be used to address any well-being issue, for the purpose of this study workers were asked to focus exclusively on hearing health. We hypothesized that, compared to a standard HCP, a participatory program would improve hearing-related attitudes and behaviors. Furthermore, we expected that workgroups with a higher level of participation would show the greatest improvements in attitudes and behaviors related to hearing loss prevention which would represent a more lasting solution.

## 2. Materials and Methods

### 2.1. Study Design and Population

We used a cluster randomized control design (Figure 1) to test study hypotheses. We engaged workers within a New England state’s Department of Transportation (DOT), specifically highway “maintainers” who are involved in road construction and maintenance activities and are regularly exposed to high noise levels. For example, during the routine task of cutting brush, we observed a mean (standard deviation) noise exposure of 86.2 (2.6) dBA TWA_8hr_ over the work shift and 92.1 (7.6) dBA L_eq_ while using the chainsaw [21]. Maintainers work out of regionally located maintenance garages. Garages were randomized as either control that received elements of the standard HCP, or HearWell that received HCP elements selected and re-designed using the HWPP. HearWell garages were further differentiated based on the presence of worker Design Teams that created interventions. Garages, rather than individuals, were assessed at each phase of enrollment and allocation.

The HearWell intervention was developed using the HWPP with a focus on hearing health. While traditional HCPs required by OSHA require several elements, given the limited scope of this project (budget and timeline) and with the direction of the Design Team, HearWell focused on the training and hearing protection components of the HCP. The other components of an HCP including audiometry, noise monitoring and control, and record-keeping, were already being addressed by the DOT’s Health and Safety Department. With respect to policy, at the time of the study, the DOT had a general policy requiring ear protection when working in an environment with sound levels that exceed 85 dB. Ear protection in the form of muffs and earplugs was issued to all maintainers. Yet, clarification on when worker noise exposures exceeded this level as well as the specific hearing protection required was at the discretion of the supervisors. The HearWell study period was from February 2016 through August 2019 (Table 1). The UConn Health Center Institutional Review Board reviewed and approved all research activities, and all participants provided informed consent.

### 2.2. Randomization of Study Sites

Maintainers worked out of 48 garage facilities spread evenly across the state. The 48 garages are classified into 8 sections or groupings based on geography. In consultation with management and for the purpose of this study, 24 garages in 4 sections were chosen based on similarity of work tasks after considering the types of roadways and typical weather events within each section (Figure 1). For randomization, researchers conducted a survey at each of these 24 garages. In light of the participatory foundations of the HWPP, the 12 garages with the highest participation in the survey (84 to 100% of on-site workers participating) were selected for randomization among the study arms (Figure 1). A random number generator was used to assign the 12 garages into groups: control, which received the standard DOT HCP elements, and HearWell, which received the HearWell HCP elements. Among the HearWell garages, the first two garages selected via random number were designated to host worker Design Teams and were classified as HearWell Design Team (Figure 1).

### 2.3. Implementation of the Healthy Workplace Participatory Program (HWPP)

Using the participatory methodology outlined by CPH-NEW [11,17], we implemented key components of the HWPP. Program elements included a 2-committee structure with a worker Design Team in two garages and a state-wide Steering Committee of representatives from Health and Safety, Operations, and Finance. Importantly, Steering Committee members supported the project by allocating paid work time for Design Team members to meet regularly and authorizing the workforce surveys at all 12 garages. Research staff served as the program facilitators, and an internal health and safety manager served as the program champion. Throughout the study period, the program champion updated the Steering Committee of the Design Team regarding activities. In addition to the study start-up, the Steering Committee was actively engaged during the design and implementation of the HearWell interventions (Table 1).

Supervisors at the two HearWell Design Team garages were each asked to recruit five workers to form the worker Design Teams. The Steering Committee and Design Teams worked together through the 7-step IDEAS process. The Design Teams each held 20 1-h meetings, beginning in March 2017 and ending in November 2018 (Table 1). In meetings 1–4, occurring every two weeks, Design Team members were trained and participated in a focus group assessing their beliefs, attitudes and opinions about hearing and noise exposure. In meetings 5–12, approximately monthly, Design Team members went through Steps 1–5 of the IDEAS process [13]. Briefly, Step 1 of the IDEAS process used a structured root-cause analysis approach to identify the factors contributing to hearing loss. Step 2 developed intervention objectives and activities to address the contributing factors. Steps 3 and 4 set and applied selection criteria to compare the strengths and weaknesses of interventions being considered by the Design Team.

The development, selection, implementation and evaluation of interventions occurred in IDEAS Steps 5–7. In collaboration with the Steering Committee, the Design Team and research staff developed interventions across four months and five Design Team meetings. The intervention included multiple components including supervisor training, a noise hazard management scheme, and a HearWell training video and presentation. Design Team members from both garages were actively engaged in drafting each intervention component. The noise hazard scheme was reviewed and adapted based on members’ suggestions. Workers suggested and edited a maintainer-specific scenario within the supervisor safety training materials to emphasize HPD use. To create the HearWell training, Design Team members reviewed numerous industry videos and presentations and assessed which elements and content they found most effective. They also discussed the relative merits of different methods to deliver the training (i.e., in-person, video, toolbox talk, and workbook).

The supervisor training was implemented in October 2018 at the 6 HearWell garages (those with and without Design Teams). Supervisors in the control garages received no additional training. In January through February of 2019, the HearWell worker training and HearWell noise hazard scheme interventions were both implemented among HearWell garages (Table 1). Workers within control garages received a standard hearing conservation training video at this time.

### 2.4. Data Collection

Survey data collection occurred at four time points across the study period: randomization, pre-intervention, training and post-intervention (Table 1). Surveys were prepared in Qualtrics™ and administered on electronic tablets. All surveys were completed during work time at the beginning or end of the work shift. A convenience sample of workers present on the day of the survey was collected at each time point. The number of garages surveyed varied by study period. In Spring 2016, the randomization survey was distributed to workers in 24 garages. In Winter 2018, the pre-intervention survey was administered among the 6 control, 4 HearWell, and 2 HearWell Design Team garages. In Winter 2019, the same 12 garages were surveyed pre- and post-training. In Summer 2019, 5 control, 4 HearWell, and 2 HearWell Design Team garages were surveyed with the 6-month post-intervention survey. One control garage was not surveyed due to staffing and work project constraints.

Each survey included items to assess demographics and job history, as well as the frequency of HPD use [21]. Workers were asked “If you use noisy tools or are in noisy areas, do you use hearing protectors (e.g., earplugs or earmuffs)?” rated on a 6–point frequency scale from rarely or never (1) to always (6).

The Design Team suggested additional survey items for the pre-intervention, training, and post-intervention surveys. A new hearing climate scale was modeled on the concept of safety climate as a way to capture attitudes about noise and hearing prevention within the workgroup. The hearing climate scale was validated prior to its use [22].

In order to address the bias inherent with asking workers to report on their own HPD use frequency, the Design Team included a question on frequency of HPD use by co-workers: “How often do your co-workers wear hearing protectors (e.g., earplugs or earmuffs) when they use noisy tools or are in noisy areas?” This used the same 6–point frequency scale as the self-assessment, from rarely or never (1) to always (6). On the pre- and post-intervention surveys, workers were also asked how often they used HPD during specific home or leisure tasks and activities. Workers used the same 6–point frequency scale or indicated that they did not perform such tasks or activities. Attitudes, beliefs, and behaviors about hearing protection were measured pre- and post-training using the NIOSH Hearing Protection Beliefs, Attitudes and Behaviors Items and Scales (BAPHLS) [23,24] with each item coded on a 5-point Likert scale from 1 (totally disagree) to 5 (totally agree). BAPHLS outcomes were summarized into 7 subscales (see results).

### 2.5. Data Analysis

An intent-to-treat approach was used in evaluating differences between control, HearWell, and HearWell-Design Team arms. The following hypotheses were tested:

**Hypothesis** **1** **(H1).**
*HearWell and HearWell Design Team garage workers will show changes in HPD beliefs, attitudes and behaviors following the HearWell training.*


**Hypothesis** **2** **(H2).**
*Compared to the control group at randomization, workers in the HearWell and HearWell Design Team garages will report an increase in HPD use frequency following supervisory training (pre-training) and 6 months following the HearWell training (post-training).*


**Hypothesis** **3** **(H3).**
*Compared to the control group at pre-training, workers in the HearWell and HearWell Design Team garages will show an increase in hearing climate (attitudes about noise and hearing prevention) 6 months following the HearWell training (post-training).*


**Hypothesis** **4** **(H4).**
*Compared to the control group at pre-intervention, workers in the HearWell and HearWell Design Team garages will show an increase in at-home HPD use (post-intervention).*


We assessed changes in HPD beliefs, attitudes and behaviors following the training by performing paired t-tests on pre- and post- values for each subscale by each independent intervention scenario group. We assessed the impact of the HearWell intervention using mixed linear modeling while accounting for repeated measures among participants. Analyses were conducted in SAS 9.4 using PROC MIXED with a REPEATED statement. A categorical variable for the survey period (randomization, pre-intervention, pre-training and post-intervention) was included in the model along with a variable to indicate intervention scenario (control, HearWell, HearWell Design Team). For each analysis, the control group at the earliest survey period served as the reference. Reported *p*-values represent the comparison of the selected survey period and intervention arm to the control group at the earliest survey period. In sensitivity analyses, stratified analyses were performed by the intervention group, with the earliest survey period serving as the reference. In addition to the main outcome variables of interest (HPD use frequency and hearing climate), a sensitivity analysis was performed by examining the difference in reported frequency of HPD use among co-workers. All analyses were conducted in SAS 9.4.

## 3. Results

### 3.1. The IDEAS Process

The two HearWell Design Teams completed the IDEAS process Steps 1 through 5 in parallel through seven meetings each. The major results are presented in Table 2. Briefly, in Step 1, Design Team members identified root causes of poor hearing health: high noise exposures due to noisy equipment and long shifts; not knowing when to wear HPD given the varying tasks and equipment used, and inconsistent supervisor policy enforcement and support for wearing HPD and reducing noise exposures (Table 2).

In Step 2, Design Team members strategized a full range of solutions without the consideration of feasibility. Solutions fell within the areas of noise reduction, policy, noise awareness, hearing culture, HPD, training, and work practices (Table 2). Researchers served as subject matter experts in hearing conservation, providing background information regarding hearing or intervention options when asked. In Steps 3 and 4, the Design Team members set and applied selection criteria to the proposed solutions to narrow and further refine the solutions under consideration. They sought solutions that were accessible to all maintainers and were also effective in protecting long-term hearing. They recognized the limited budget for the solutions as well as potential resources including trainers as well as the knowledge of the Design Team members themselves. In the end, Design Team members had crafted a set of solutions for the Steering Committee to consider implementing.

### 3.2. HearWell Intervention Components

In IDEAS Step 5, the Steering Committee and Design Team worked together to select, adapt and create three main HearWell intervention components; supervisor training, a noise hazard management scheme, and the HearWell training (Table 2).

The goal of the supervisor training was to increase leadership support of hearing and HPD use and ultimately create a safe and consistent environment supportive of hearing health. The Foundations for Safety Leadership (FSL) Training was identified as a relevant component as its goal is to teach construction supervisors leadership skills to strengthen workplace safety climate and reduce adverse safety outcomes [25]. With permission from FSL creators, the program materials were adapted with formatting and minor content editing for garage supervisors and crew leaders. In addition, we created and implemented a scenario specific to transportation maintenance noise hazards during brush cutting. The FSL training was a 2.5 h, instructor-led training that included videos along with a workbook for participants.

The goal of the noise hazard management scheme was to easily identify noise levels of individual equipment models and indicate the required HPD for each. The Design Team, Steering Committee and researchers developed a color-coded noise hazard scheme that allows workers to recognize equipment and task noise levels and choose the appropriate HPD [21]. Briefly, the noise hazard scheme identified noise level ranges (red, above 105 dBA; orange, 90–105 dBA; and yellow, 85–90 dBA) and the corresponding HPD required (red, muffs and plugs; orange, muffs or plugs; yellow, muffs or plugs or ear caps). The noise hazard scheme was developed using a participatory and iterative process. Ultimately, the workers opted for a scheme that incorporated the noise levels of each piece of equipment, regardless of exposure time and the selection of HPD based on what was already available to them [20]. The scheme complemented and further clarified the existing noise policy, requiring HPD use when noise exposures exceeded 85 dB.

The goal of the HearWell training was to educate workers (maintainers and supervisors) on the noise hazard scheme as well as the content required by OSHA and other information suggested by Design Team members. Topics required by OSHA (CFR 1910.95) were incorporated into the training including: (a) the effects of noise on hearing; (b) the purpose of hearing protectors, the advantages, disadvantages, and attenuation of various types and instructions on hearing protectors selection, fit, use, and care; and (c) the purpose of audiometric testing, and an explanation of the test procedures. In addition to training on the noise hazard scheme, Design Team additions included: (a) ways to reduce noise exposure (optimizing distance from noise source, trading off noisy jobs); (b) assessing noise levels (a noise thermometer mapping noise levels of common tools/equipment; the “shouting test” to assess if noise levels are beyond 85 dBA; and (c) ways that co-workers can support HPD use and practices to reduce noise exposures. The Design Teams chose a video format to provide a consistent experience and allow for easier rollout. Following the video, workers suggested a demonstration of proper HPD use including earplug insertion and muff adjustment as well as time for questions and answers.

### 3.3. Survey Results

A total of 271, 163, 140 and 95 workers were surveyed at randomization, pre-intervention, pre-training, and post-intervention, respectively. Survey participation at randomization averaged 81% among the 12 garages. Among the randomized control and HearWell garages, high participation was maintained for the pre-intervention, pre-training, and post-intervention surveys (means 94%, 96%, 86%, respectively). A total of 288 participants were surveyed with 131 (46%) providing one survey, 78 (27%) providing 2 surveys, 52 (18%) providing 3 surveys and 27 (9%) providing 4 surveys. Survey respondents were predominantly male, White, and maintainers, with a mean age of 44 years and a mean of 11 years in the current job (Table 3). All characteristics remained similar across surveys (Table 3). Among surveyed workers, 20 changed garages across the study period; 11 workers moved from the excluded garages to the intervention garages, 4 workers had no change in the intervention arm, 2 workers changed from control to HearWell, 2 workers changed from HearWell Design Team to control, and 1 worker changed from HearWell to control.

Eighteen supervisors and crew leaders attended the supervisors’ training. In total 73 workers received the HearWell training, and 72 workers received the control training. Participation in the pre-training survey was high: 99% (*n* = 72) of HearWell garage workers and 94% (*n* = 68) of control garage workers. The majority (84%) of workers who received the training were maintainers; three workers did not indicate a job title (Table 3).

Table 4 indicates the pre- and post-training beliefs, attitudes and values for workers by intervention arm (Hypothesis 1). There were significant changes in some, but not all of the BAPHLS subscales for the HearWell trained workers in both the HearWell and HearWell Design Team arms. Specifically, maintainers who had received the HearWell training perceived lower barriers to HPD use afterward (Table 4). Workers in the HearWell intervention garages also reported increased self-efficacy in HPD use and increased social norms around hearing protection (Table 4). Workers receiving the control training did not have meaningful changes in beliefs or attitudes about hearing protection, hearing and noise (Table 4).

To assess intervention impact, we evaluated the frequency of HPD use for each survey by intervention arm (Hypothesis 2) using a mixed-effects model to account for repeated measures of participants across surveys. Figure 2 presents the reported frequency of HPD use by maintainers for each survey by control, HearWell and HearWell Design Team intervention arms. Following randomization, the HearWell Design Team garage workers reported meaningful increased HPD use frequency after the Design Teams were formed (pre-intervention) with a continued increase in HPD use up to 6-months post-intervention. Although the HearWell garages workers also reported an increase in HPD use frequency, there was no significant statistical difference as compared to the control except for at randomization where workers in the HearWell garages reported a significantly lower frequency of HPD use as compared to the control. In sensitivity analyses, the scale point increase in HPD use frequency by intervention arm, from randomization to 6-months post-intervention, was calculated as 0.98 (95% CI: 0.30 to 1.67) for HearWell; 0.92 (95% CI: −0.10 to 1.93) for HearWell Design Team; and 0.43 (95% CI: −0.10 to 0.96) for control garage workers.

In a sensitivity analysis, HPD use frequency by co-workers was examined as an alternative to self-report (data not shown). Co-worker HPD use frequency was consistently lower than self-reported HPD use frequency, with a mean (SD) of 4.13 (1.19); 4.37 (1.46); and 5.0 (0.94) for control, HearWell and HearWell Design Team garage workers, respectively at post-intervention. Yet, similar trends were observed over time and by the intervention arm. The scale point increase in HPD use frequency by intervention arm from pre-intervention (when the co-worker HPD use frequency variable was first introduced) to 6-months post-intervention was calculated as 0.74 (95% CI: 0.12 to 1.29) for HearWell, 0.75 (95% CI: −0.01 to 1.50) for HearWell Design Team, and 0.55 (95% CI: 0.03 to 1.06) for control garage workers.

We also assessed hearing climate at pre-training and post-intervention by intervention arm (Hypothesis 3). Using linear mixed models with the control at pre-training as a reference, we found no meaningful difference in hearing climate by intervention arm at pre-training (Table 5). However, as compared to control at pre-training, meaningful improvements in hearing climate were observed across all intervention arms at 6-months post-intervention. Furthermore, the HearWell Design Team garages showed the largest improvement, up 24% from pre-training to post-intervention, and the highest value at the end of the study (Table 5).

For Hypothesis 4, with the exception of one activity (snow blowing), there were no meaningful differences in at-home HPD use among study arms, at pre-intervention (Table 6). The frequency of at-home HPD use increased for each intervention arm at 6-months post-intervention with the exception of chainsaw use and leaf blowing, which had the highest frequency of use at pre-intervention.

## 4. Discussion

The HearWell intervention created by the worker Design Teams and Steering Committee was comprised of multiple components including supervisor leadership training, noise hazard management scheme, and customized hearing prevention training. These intervention components built upon an existing policy that directed workers when to wear HPD, yet offered little guidance on identifying high noise levels or selecting the correct HPD. The goals of the intervention were to increase HPD use; improve the hearing climate in the workplace, including supervisor and co-worker support for HPD use; and reduce ambient noise levels. While all three intervention arms (control, HearWell, HearWell Design Team) showed increases in HPD use over the study period, the only statistically significant increase was in the HearWell Design Team garage workers, who also reported the highest HPD use frequency 6 months after the intervention. Both HearWell and HearWell Design Team workers reported statistically significant increases in some beliefs and attitudes about hearing loss and hearing protection following the HearWell training. Interestingly, all three intervention arms showed an improvement in hearing climate as well as increases in at-home HPD use.

The training was shown to be an effective means of improving HPD use [26,27,28]. As was observed in HearWell, tailored training appears to be most effective in increasing HPD use [28]. Among construction workers, both train-the-trainer and expert-based 1-hour oral presentations with physical demonstration increased knowledge and self-efficacy and decreased barriers with respect to hearing and HPD [26]. Multi-component approaches, including hearing loss prevention training, refresher training along with noise-level indicators have been successful in increasing HPD use among construction workers [29].

### 4.1. Defining Integration for Total Worker Health

In addition to its ability to improve hearing-related outcomes, the success of HearWell can also be assessed in its ability to achieve a TWH approach. TWH is defined by the NIOSH as “policies, programs and practices that integrate protection from work-related safety and health hazards with promotion of injury and illness prevention efforts to advance worker well-being” [30]. While the fundamental approach in TWH programs is to adopt integrated approaches for promoting worker health and well-being, there remain different perspectives on what defining elements should be included [9,11,31,32]. As practiced in the HWPP introduced by CPH-NEW, there are four criteria that define the content and process of integrated TWH interventions: (i) emphasis on work interventions that prioritize workplace contributors to poor well-being; (ii) assessment of both work and non-work risk factors; (iii) participatory engagement of workers; (iv) coordination of goals and activities across programs that support worker well-being [11]. These four criteria provide a useful context to assess the HearWell intervention in terms of its TWH approach.

#### 4.1.1. Interventions That Prioritize Workplace Contributors to Poor Well-Being

A comprehensive TWH approach would prioritize workplace changes to promote well-being and can be expected to be consistent with the industrial hygiene hierarchy of controls with respect to noise exposure. This prioritizes eliminating noise exposure followed by buying quiet equipment [33] and tools, controlling the sources of noise, shortening workers’ noise exposure time, and as a last resort use of HPD [34]. Although both the Design Teams and Steering Committee discussed initiatives that consider noise level when purchasing equipment, this was not prioritized due to cost and complexities in state purchasing cited as a large barrier. Rather, the HearWell program focused on reducing noise exposures through increased HPD use and education. This approach was desirable due to the low cost, available training resources and ability to impact all workers.

A TWH approach to hearing preservation also considers workplace policies, programs, and work organizations as well. HearWell addressed workplace hearing health and noise culture through the supervisor training in safety leadership, adapted from the FSL training for construction leaders [25]. The FSL training was shown to increase supervisor leadership skills, yet to date, changes in safety climate and reported supervisor practices by the crew have not been observed in prior evaluations of the training [35].

Additional facets of work organization addressed through the HearWell training included co-workers’ support for reducing noise exposures. Design Team member suggestions for increasing co-worker support for HPD use included (a) keeping extra plugs in the truck for co-workers who forget them; (b) reminding and encouraging co-workers to use HPD, especially those new to the job; and (c) before starting up a noisy piece of equipment, warning co-workers to check that they are wearing HPD.

Administrative controls to reduce noise exposure were also part of HearWell interventions. Job rotation or switching off on noisy equipment such as the chainsaw was emphasized within the HearWell training. In addition, within Design Team garages, the noise hazard color scheme was used when assigning morning work orders. Supervisors and crew leaders added the color code corresponding to the noise level and required HPD to be added to the work order as a way to emphasize that HPD was required.

Overall, HearWell addressed workplace contributors to hearing health, yet the approaches that were adopted could have been further improved by prioritizing the hierarchy of controls with respect to noise exposure; namely noise exposure elimination or reduction at the sources over the use of HPD. Perhaps after a longer period of collaboration, members of the Design Team would no longer rule out the selection of quiet equipment due to feasibility concerns. Nonetheless, the reliance on HPD use to control noise exposure, while less desirable than engineering controls, is relatively commonplace and has been implicated as a driver of the high burden of hearing loss among noise-exposed workers [4]. HPD use can be improved upon by performing fit-testing or by choosing HPD that incorporates advanced attenuation technologies and/or features that promote communication between coworkers.

#### 4.1.2. Assessment of Both Work and Non-Work Factors

A comprehensive TWH approach to promoting hearing health would also consider work and non-work risk factors for hearing loss. In addition to workplace noise exposures, non-work exposure to noise at home and in the community can also put workers at risk for hearing loss. Among maintainers within this study, noisy tasks such as lawn mowing or leaf blowing may occur concurrently at work and home increasing daily noise exposure levels. As part of the HearWell program, workers addressed at-home noise exposure by encouraging protective behavior outside of work and surveying at-home HPD use. Interestingly, we observed an increase in at-home HPD use for the majority of tasks over time regardless of study arms. At-home HPD use for chainsaw use was high at pre-intervention and changed little post-intervention. However, prior to the pre-intervention survey, researchers conducted in-depth noise survey monitoring during tree removal [21], which includes chainsaw use, which may have contributed to the increase in at-home HPD use during this task at this time. Nevertheless, it is unclear why all arms showed an increase in at-home HPD use. It is possible that given the low frequency of at-home HPD use at pre-intervention, raising worker awareness about the benefits of hearing protection in general may have helped workers to reconsider HPD use at both work and home.

A comprehensive, TWH approach to hearing health considers work and non-work exposure to ototoxic chemicals promotes healthful living and monitoring activities including regular audiometric exams [36]. HearWell can be improved through a broader recognition of hearing health risk factors.

#### 4.1.3. Participatory Engagement

Participatory engagement is the cornerstone of a TWH approach [11]. A participatory TWH approach is at the forefront of the HWPP and the IDEAS process, and therefore participatory engagement was central to the development, implementation and assessment of HearWell. The participatory process of HearWell was achieved through the direct engagement of maintainer representatives in the HearWell Design Team garages as well as the key decision-makers that made up the Steering Committee.

Participatory engagement appeared to have played a key role in the success of HearWell. HPD use frequency was highest across time and showed a statistically significant improvement for the HearWell Design teams compared to the control garages. The benefits of employing a participatory process when designing interventions are known to be multifold [10] and provide insights into the success of HearWell. Employee engagement in problem identification and solution development can help to identify issues that are most relevant to the workers, create interventions that are contextually appropriate, and identify a broader range of contributing factors such as work organization that can significantly affect workers [10]. This was definitely the case for HearWell where the Design Teams’ intervention approach was multi-faceted. The intervention incorporated knowledge and behavior changes as well as work organization factors and was inclusive of supervisors as well as line workers. Training elements were tailored to the maintainers’ job with realistic scenarios and tasks and also included videos in which peers described hearing loss consequences and HPD use challenges. Furthermore, the training purposefully addressed the noise and hearing concepts identified by the Design Team members as most relevant, including ways to reduce and assess noise exposure as well as the importance of co-worker support for noise reduction.

While both HearWell and HearWell Design Team garage members benefited from the HearWell interventions, direct employee engagement in TWH efforts promotes employee self-efficacy [10] which may have impacted HearWell garages with Design Teams more so than those garages without Design Teams. In fact, the HearWell Design Team members considered themselves “hearing ambassadors” and resources to their garages, which likely influenced co-worker HPD use frequency. The success of employee engagement is consistent with best practices for HCP where it was suggested that employee education and motivation are the keys to a successful HCP [37,38]. Employee engagement also benefits program sustainability [10]. While the long-term effects of HearWell are unknown, increased HPD use 6-months post-intervention is a promising indicator of program longevity. In addition to identifying barriers and risk factors for hearing health and crafting solutions, employee engagement should be a key component in continued program evaluation and improvement for increased program sustainability.

#### 4.1.4. Coordination of Programs and Activities

A TWH approach involves the coordination of programs and activities that support worker well-being which can occur across a spectrum ranging from parallel programs with cross-promotion of specific activities to programs that are integrated through the common goal of promoting hearing health. Therefore, coordination to support hearing health may encompass a broad range of programs, policies and departments that support the reduction of noise exposure as well as the promotion of behaviors supporting hearing health.

Through the Steering Committee, the HearWell program advanced coordination of noise reduction strategies with other ongoing initiatives. The Steering Committee included representatives from finance as well as operations, and meetings focused on topics that included the importance of considering noise levels when buying new equipment, the need for a variety of HPDs for workforce fit and comfort, staffing needs, and best practices for limiting noise exposures. Notably, the HearWell interventions were limited by the small budget provided by the research staff for materials that included worksheets, posters, video production software and sample HPD. Higher expenditures for training and HPD fit-testing have been associated with reduced hearing loss documented through standard threshold shifts [39] and might have increased program effectiveness here. An integrated TWH approach to hearing health would also include coordination with health promotion and medical screening.

### 4.2. Limitations

The HearWell study had a number of noteworthy limitations. Ideally, the goal of an HCP is to preserve hearing health and reduce noise exposures. Yet, for the purpose of the HearWell pilot program, we did not have access to audiometry nor comprehensive noise exposure data for either motivating workers or evaluating HearWell effectiveness. Rather, HearWell relied on self-report measures, namely frequency of HPD use as well as hearing climate, to assess intervention effectiveness. While there is an inherent bias in self-report measures, in a sensitivity analysis we observed similar trends when workers were asked to report on co-workers HPD use frequency. Furthermore, small sample sizes may have limited our ability to detect differences in the groups. While garages were randomized at the study start, small sample sizes may have contributed to the unequal distribution of HPD use frequency, with the HearWell garages with the lowest HPD use at randomization. Some workers also switched between garages over the course of the study. Those transfers might have led to misclassification with regard to the intervention dose actually received. While the intention-to-treat analytic approach is appropriate for control of selection bias, misclassification could have biased these findings towards the null value, further limiting our ability to observe statistically significant associations. Yet despite these limitations, the HearWell pilot intervention showed promising results in improving HPD use through a participatory TWH approach to hearing conservation.

### 4.3. Implications for the Future of Work

The work performed by maintainers has and can be expected to change in the future due to myriad factors linked to climate change. In addition to increased outdoor heat exposures due to climate change-related heat increases, maintainers are also affected by the increasing frequency, duration and severity of extreme weather events caused by climate change [40]. With respect to HearWell, maintainers recognized that long shifts with extended noise exposure often resulting from storm work are a root cause of hearing loss. This suggests that regular involvement by the maintainer workforce in designing interventions as part of a TWH program will be necessary to address changes in their future work that can impact their health and well-being.

Consistent with the future of work trends across the workforce in general, maintainers are a multi-generational and aging workforce. The participatory methods employed in HearWell allowed for a range of worker attitudes, beliefs and needs to be heard and addressed. When the Design Team engaged in the IDEAS processes including root causes analysis, this consensus-building process was able to incorporate the diversity of concerns present across the entire workforce. Importantly, the participatory design process and Design Team experience overall promoted skill development in the areas of interpersonal communication and group problem-solving, both of which are recognized as core competencies required for a successful future workforce [12].

## 5. Conclusions

The results of the present study indicate that the well-established occupational safety and health area of noise and hearing conservation can benefit from adopting a TWH approach. Employee engagement in hearing health promotion efforts was found to be an effective way of promoting behaviors that support hearing protection and instituting organizational changes in support of hearing conservation. Although employees benefitted from receiving the same interventions that were developed by employees elsewhere, maximum program impact was found at those sites in which employee design teams actively engaged in intervention design efforts. Thus, the present findings suggest that future hearing conservation programs should leverage employee participation for maximal effectiveness.

## Figures and Tables

**Figure 1 ijerph-18-09529-f001:**
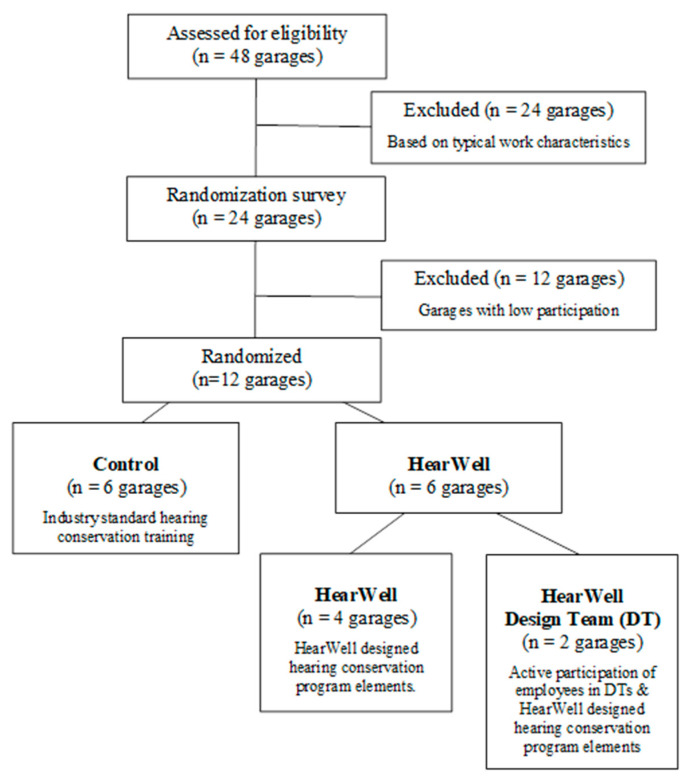
Flow diagram of the allocation of Department of Transportation maintenance garages into study arms.

**Figure 2 ijerph-18-09529-f002:**
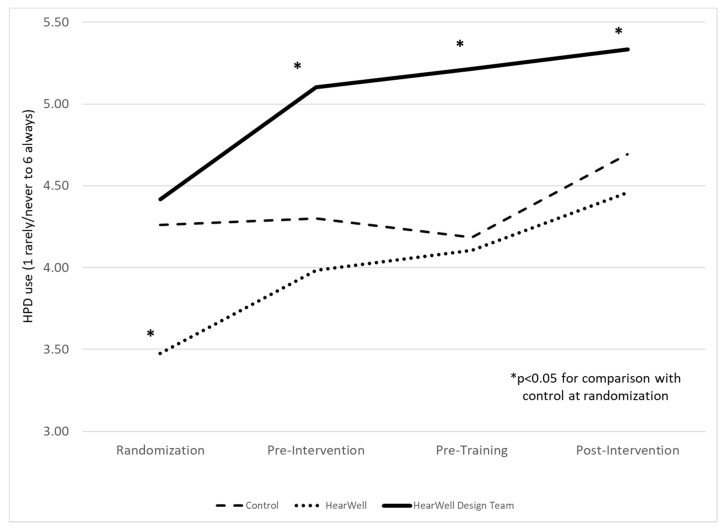
Reported frequency of HPD use across surveys by intervention arm among transportation maintainers. Mixed linear models accounting for repeated measures were used to evaluate difference between survey periods and intervention arms. Statistical significance (*p* < 0.05) is presented for the comparison between each time period, by intervention arm with respect to the control values at randomization.

**Table 1 ijerph-18-09529-t001:** HearWell Study period timeline and major activities.

Year (s)	Month (s)	Participants	Activities
2016	Feb	SC	HearWell start-up
2016	Apr–May	Full Workforce ^1^	Eligibility assessment, randomization survey
2017	Mar–May	DT ^2^	Team formation and training
2017	May–Oct	DT ^2^	IDEAS Steps 1–5
2017/2018	Dec–Jan	Workforce ^3^	Pre-intervention survey
2018	Apr–Nov	SC, DT ^2^	Intervention development, IDEAS Steps 5–7
2018	Oct	Supervisors ^4^	Supervisor training
2019	Jan–Feb	Workforce ^3^	HearWell or control training; Pre- and post-training surveys
2019	Jun–Aug	Workforce ^3^	Post-intervention survey

SC, Steering Committee; DT, Design Team; IDEAS, Intervention Design and Analysis Scorecard. ^1^ N = 24 garages. ^2^ N = 2 HearWell garages. ^3^ N = 12 garages (control, HearWell, HearWell Design Team); ^4^ N = 6 garages (HearWell, HearWell Design Team).

**Table 2 ijerph-18-09529-t002:** Summary of selected IDEAS Toolkit Step results from HearWell Design Teams addressing hearing health at two transportation maintenance garages.

Activity	Outcome
Identify Health and Safety Problems; Contributing Factors Attributed to Hearing Loss (IDEAS Step 1)	
	High noise exposures from loud equipment.
	Extended noise exposures due to long shifts and specialization of tasks requiring no variation in noise exposures.
	HPD: hot, uncomfortable, limited options, unsure of proper use/replacement.
	Knowledge gap: unsure of when to use HPD, noise levels of equipment.
	Safety climate: lack of supervisor and coworker support for wearing HPD and reducing noise exposures.
Brainstorm Solution Activities (IDEAS Step 2)	
Noise reduction	Buy less noisy equipment.
	Retrofit mowers and vehicles to reduce in-cab noise exposures.
Policy	Upgraded, written policy specific to hearing protection, with detail on equipment/situations that require HPD.
	Enforcement for policy should including escalation of consequences (e.g., verbal warning for 1st offense, then written warning) and a transition period to counsel workers.
Noise Awareness	Color-coded system to easily identify noise levels.
	Stickers on equipment to identify noise levels.
Hearing Culture	Increase coworker HPD use support and awareness of bystander noise exposures.
	Support of HPD use to create consistency in hearing culture across garages.
	Hearing and noise awareness campaign (e.g., HearWell week/day/month) with hearing/noise theme tailgate talks and training.
	Posters, information on hearing, HPD use and dangerous noise levels.
HPD	More HPD options to increase comfort and fit.
	HPD allows for communication and warning signal awareness.
Training	Supervisor training on HPD policy and enforcement.
	All worker in-depth training with Design Team recommended content.
	Add training on noise and HPD use to existing tailgates (e.g., a tailgate on chainsaw blade safety should include HPD use).
	Design Team members can serve as hearing ambassadors to assist in training and best practice demonstration.
Work practices	Use a safety spotter to relay warning signals when over protection from HPD may diminish ability to hear critical noises.
	Daily work orders should specify noise level and HPD required based on equipment or task assigned.
HearWell Intervention Components (IDEAS Step 5)	
Supervisor Training	Supervisor training on safety leadership skills including HPD use support.
Noise Hazard Scheme	Color-coded system to identify noise level and required HPD based on task or equipment.
HearWell Training	A 30-min video with OSHA-required and HearWell Design Team identified topics, including job-specific scenarios and context delivered in-person with a hands-on demonstration of PPE use.

**Table 3 ijerph-18-09529-t003:** Population characteristics across survey periods at 12 Department of Transportation garages: Data from worker surveys.

	Randomization	Pre-Intervention	Pre-Training	Post-Intervention
	*n* (%) or Mean (SD)	*n* (%) or Mean (SD)	*n* (%) or Mean (SD)	*n* (%) or Mean (SD)
Male	254 (97)	156 (97)	136 (99)	91 (96)
Racial category				
White	194 (74)	113 (69)	97 (70)	67 (70)
Black	34 (13)	19 (11)	19 (14)	14 (15)
American Indian/Alaska Native; Asian; multiple or unknown races	34 (13)	26 (16)	22 (16)	14 (15)
Age (years)	44 (11)	44 (10)	44 (10)	44 (10)
Job tenure (years)	11 (6)	12 (11)	10 (9)	11 (10)
Job Title				
Maintainer	227 (87)	135 (85)	110 (84)	77 (84)
Supervisor	36 (13)	24 (15)	21 (16)	15 (16)

**Table 4 ijerph-18-09529-t004:** Hearing protection belief, attitudes and behaviors (BAPHLS) sub-scales pre- and post-training by intervention arm. Mean (standard deviation) of values from 5-point Likert scale from totally disagree (1) to totally agree (5). Values are bolded to indicate statistically significant differences. *p*-values are from paired *t*-test between pre- and post-values.

	Control			HearWell			HearWell Design Team		
	Pre	Post	*p*-Value	Pre	Post	*p*-Value	Pre	Post	*p*-Value
Barriers to preventive action (reverse coded)	3.33 (0.65)	3.32 (0.63)	0.89	**3.38 (0.60)**	**3.63 (0.79)**	**0.01**	**3.36 (0.63)**	**3.65 (0.65)**	**<0.0001**
Self-efficacy	4.00 (0.56)	4.10 (0.54)	0.12	**3.89 (0.70)**	**4.13 (0.69)**	**0.04**	4.06 (0.52)	4.17 (0.54)	0.16
HPD behavior intent	4.07 (0.67)	4.12 (0.65)	0.29	3.93 (0.71)	4.01 (0.65)	0.43	4.33 (0.52)	4.27 (0.42)	0.33
Social norms	3.89 (0.52)	3.87 (0.52)	0.66	**3.64 (0.63)**	**3.84 (0.65)**	**0.001**	4.06 (0.36)	4.07 (0.38)	0.87
Susceptibility to hearing loss	4.13 (0.54)	4.13 (0.55)	0.89	4.13 (0.52)	4.19 (0.61)	0.39	**4.14 (0.51)**	**4.27 (0.41)**	0.04
Benefits of protective action	4.35 (0.53)	4.46 (0.57)	0.12	4.30 (0.61)	4.39 (0.58)	0.39	4.34 (0.51)	4.39 (0.53)	0.64
Severity of hearing loss consequences	4.51 (0.62)	4.42 (0.65)	0.24	4.46 (0.69)	4.35 (0.66)	0.27	4.23 (0.68)	4.27 (0.71)	0.73

**Table 5 ijerph-18-09529-t005:** Hearing climate by intervention arm pre-training and post-intervention. *p*-values represent the statistical significance as compared to the control at pre-training using a mixed linear model.

	Pre-Training		Post-Intervention	
	Mean (SD)	*p*-Value	Mean (SD)	*p*-Value
Control	3.26 (0.76)	Reference	3.89 (0.79)	0.0003
HearWell	3.07 (0.77)	0.24	3.74 (0.92)	0.005
HearWell Design Team	3.18 (0.65)	0.65	3.94 (0.77)	0.001

**Table 6 ijerph-18-09529-t006:** Self-reported at-home hearing protection device use frequency (1 rarely/never to 6 always) by task and intervention arm. *p*-values represent the statistical significance as compared to the control at pre-intervention using a mixed linear model.

	Pre-Intervention				Post-Intervention			
	*n*	Mean	(SD)	*p*-Value	*n*	Mean	(SD)	*p*-Value
**Mowing**								
Control	67	2.52	(1.99)	Ref	32	4.06	(1.29)	0.0004
HearWell	55	2.80	(1.98)	0.41	26	3.96	(1.66)	0.002
HearWell Design Team	29	2.93	(2.02)	0.33	17	3.82	(1.70)	0.01
**Chainsaw**								
Control	66	3.41	(2.10)	Ref	25	3.64	(1.66)	0.62
HearWell	58	3.36	(2.05)	0.90	24	3.38	(1.88)	0.94
HearWell Design Team	27	4.04	(1.97)	0.18	14	3.57	(1.79)	0.78
**Power tools**								
Control	69	2.87	(1.87)	Ref	28	3.89	(1.17)	0.01
HearWell	57	3.04	(1.79)	0.59	30	3.57	(1.61)	0.07
HearWell Design Team	28	3.54	(1.86)	0.09	17	3.88	(1.32)	0.04
**Leaf blowing**								
Control	68	3.18	(2.09)	Ref	28	4.00	(1.41)	0.07
HearWell	58	3.26	(1.99)	0.81	27	3.70	(1.77)	0.24
HearWell Design Team	28	3.75	(2.01)	0.23	16	3.69	(1.85)	0.35
**Plowing**								
Control	52	1.37	(1.07)	Ref	34	5.09	(0.87)	<0.0001
HearWell	48	1.38	(1.14)	0.97	34	4.71	(1.47)	<0.0001
HearWell Design Team	23	1.39	(1.12)	0.93	19	4.84	(1.21)	<0.0001
**Snow blowing**								
Control	56	1.75	(1.48)	Ref	30	4.83	(1.29)	<0.0001
HearWell	49	2.24	(1.81)	0.12	30	4.87	(1.46)	<0.0001
HearWell Design Team	26	2.73	(2.07)	0.02	18	4.89	(1.13)	<0.0001
**Motorized recreational vehicles**								
Control	55	1.62	(1.30)	Ref	35	5.14	(0.60)	<0.0001
HearWell	52	1.42	(0.98)	0.42	33	4.48	(1.48)	<0.0001
HearWell Design Team	25	2.08	(1.87)	0.13	18	5.00	(0.84)	<0.0001
**Music concerts and racetracks**								
Control	54	1.78	(1.57)	Ref	35	4.74	(1.01)	<0.0001
HearWell	52	1.77	(1.46)	0.98	34	4.50	(1.01)	<0.0001
HearWell Design Team	24	2.29	(1.94)	0.17	18	4.33	(1.41)	<0.0001

## Data Availability

The data presented in this study are available on request from the corresponding author. The data are not publicly available due to ethical restrictions.

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
