# Peer review of "Evaluation of the HearWell Pilot Program: A Participatory Total Worker Health® Approach to Hearing Conservation"

_ijerph, 2021, doi:10.3390/ijerph18189529_

Round 1
Reviewer 1 Report
The paper, entitled "Evaluation of the HearWell Pilot Program: A participatory Total Worker Health® approach to hearing conservation", describes a long and carefully conducted study on hearing conservation.
Education about the effects of noise on hearing and hearing conservation is extremely important.
This paper examines the effect of the HearWell as the proposed method on these issues, and the effect is also statistically significant. However, since it is not written which test method was used, it is not possible to discuss the statistical significance of the difference, significance of content, and scientific soundness. Please describe the test method and test statistic.
Reviewer 2 Report
The manuscript covers a very important aspect of protecting workers' hearing and has been carefully prepared. Nevertheless, the following comments should be made to remedy the deficiencies noted:
- The value of the manuscript would be greater if values of the noise parameters to which the workers were exposed were provided.This deficiency was noticed in the section on limitations, however, it is worth considering the possibility of characterizing examples of situations in which noise was generated. Such information would enable the reader to broaden their knowledge of the problem.
- It is recommended to expand the information on the colors used to indicate noise situations.There was no explanation about the individual colors that were used to mark situations related to exposure to noise.What were the criteria for marking certain situations with specific colors?What noise parameter was used and what were the limit values?
- There was a lack of information whether the selection of the used hearing protectors was carried out, taking into account the noise parameters? Please comment on this point.
- The manuscript should explain the rules for using HPDs.It is not known at what noise sound pressure level it was an obligation. At what noise sound pressure level the employer provided hearing protectors, but the use of HPDs was only recommended, but not compulsory? How did the principles of using HPDs affect the way the frequency of wearing of hearing protectors is measured?
- It is necessary to clarify what is meant by "hearing climate" (Line: 217).
- It is worth commenting on what may be hidden under the entry in Table 2: " Use a safety spotter when HPD use may compromise safety.” Mainly what kind of detector is meant and in what situations the use of HPD may endanger safety.
- Inconsistent numbering of hypotheses (3a, 3b) (Line: 352)with the previously used (3) requires correction.
Round 2
Reviewer 1 Report
- Statistical tests in Table 4
It is the multiple tests (the t-test is used repeatedly). In this case, we need to change the interpretation of the p-value; please correct the p-value by using the Bonferroni method or the Sidak method.
